# Changes in Adolescents’ Psychosocial Functioning and Well-Being as a Consequence of Long-Term COVID-19 Restrictions

**DOI:** 10.3390/ijerph18168755

**Published:** 2021-08-19

**Authors:** Nóra Kerekes, Kourosh Bador, Anis Sfendla, Mohjat Belaatar, Abdennour El Mzadi, Vladimir Jovic, Rade Damjanovic, Maria Erlandsson, Hang Thi Minh Nguyen, Nguyet Thi Anh Nguyen, Scott F. Ulberg, Rachael H. Kuch-Cecconi, Zsuzsa Szombathyne Meszaros, Dejan Stevanovic, Meftaha Senhaji, Britt Hedman Ahlström, Btissame Zouini

**Affiliations:** 1Department of Health Sciences, University West, 46186 Trollhättan, Sweden; maria.erlandsson@hv.se (M.E.); britt.hedman.ahlstrom@hv.se (B.H.A.); 2AGERA KBT AB, 41138 Gothenburg, Sweden; kourosh@meshe.se; 3High Institute of Nursing Professions and Health Techniques, BP 57, Errachidia 52000, Morocco; anis.sfendla@gmail.com; 4Department of Biology, Faculty of Sciences, Abdelmalek Essaâdi University, Avenue de Sebta, Mhannech II, Tetouan 93002, Morocco; mohjat.belaatar@etu.uae.ac.ma (M.B.); abdennour.elmzadi@etu.uae.ac.ma (A.E.M.); msenhaji@uae.ac.ma (M.S.); btissamezouini@gmail.com (B.Z.); 5Department of Biology, Faculty of Sciences and Techniques, Moulay Ismail University, BP 509, Boutalamine, Errachidia 52000, Morocco; 6Department of Psychiatry, Clinic for Neurology and Psychiatry for Children and Youth, 11000 Belgrade, Serbia; vladimirjvc@gmail.com (V.J.); stevanovic.dejan79@gmail.com (D.S.); 7Department of Social Sciences, Faculty of Education in Sombor, University of Novi Sad, 25000 Sombor, Serbia; radedamjanovic@yahoo.com; 8Department of Clinical Psychology, Faculty of Psychology, University of Social Sciences and Humanities, VNU, Hanoi 100000, Vietnam; ntmhang@vnu.edu.vn; 9Department of Social Work with Children and Family, Faculty of Social Work, Hanoi National University of Education, Hanoi 100000, Vietnam; nguyetnta2512@gmail.com; 10Department of Psychiatry and Behavioral Sciences, SUNY Upstate Medical University, Syracuse, NY 13210, USA; ulbergs@upstate.edu (S.F.U.); KuchR@upstate.edu (R.H.K.-C.); meszaroz@upstate.edu (Z.S.M.)

**Keywords:** adolescents, COVID-19, exercise, gender, mental health, norm-breaking behaviors, psychosocial functioning, substance use, stress, victimization

## Abstract

This work studied self-reports from adolescents on how the COVID-19 pandemic has changed their behaviors, relationships, mood, and victimization. Data collection was conducted between September 2020 and February 2021 in five countries (Sweden, the USA, Serbia, Morocco, and Vietnam). In total, 5114 high school students (aged 15 to 19 years, 61.8% females) responded to our electronic survey. A substantial proportion of students reported decreased time being outside (41.7%), meeting friends in real life (59.4%), and school performance (30.7%), while reporting increased time to do things they did not have time for before (49.3%) and using social media to stay connected (44.9%). One third of the adolescents increased exercise and felt that they have more control over their life. Only a small proportion of adolescents reported substance use, norm-breaking behaviors, or victimization. The overall COVID-19 impact on adolescent life was gender-specific: we found a stronger negative impact on female students. The results indicated that the majority of adolescents could adapt to the dramatic changes in their environment. However, healthcare institutions, municipalities, schools, and social services could benefit from the findings of this study in their work to meet the needs of those young people who signaled worsened psychosocial functioning, increased stress, and victimization.

## 1. Introduction

The new SARS-CoV-2 (Coronavirus), a life-threating global infectious respiratory disease, spread rapidly around the world from the end of 2019 to the beginning of 2020, leading to the COVID-19 pandemic. Countries across the world ordered partial or total lockdowns for their societies, as the seriousness of the COVID-19 pandemic became evident. Children and adolescents were not considered at high risk for COVID-19 infection, and it was recognized that they had a milder disease course and a better prognosis than adults [1]. However, countries with middle- and low-income populations observed more severe causalities, and this impacted children and adolescents [2]. This standpoint may have caused a delay in attention being paid to children’s and adolescents’ well-being during the pandemic.

Considering that children and adolescents were not those at highest risk from the pandemic, some may question whether one should investigate the impact of COVID-19 on adolescents’ mental health and well-being at all. Adolescence is a vulnerable period during human development. During this time, active physical (growth), neurobiological (pruning), physiological (hormonal), and psychological (self-image, social autonomy, and increased impulse control) changes occur [3,4,5]. These intensive changes make individuals more vulnerable to environmental influences. Extreme environmental factors, such as a pandemic and its restrictions, can be considered developmental risks, which may have a profound psychological impact on adolescents’ long-term development, health, and well-being.

During the COVID-19 pandemic, restrictions have been introduced at varying rates and to different extents in different countries around the world. Even if restrictions are similar across several countries, the respective socioeconomic resources of those countries can differ greatly, which can influence how much of a negative societal impact the restrictions have. It is therefore highly important to form an overall understanding of the global impact of the COVID-19 pandemic restrictions on adolescents’ psychosocial functioning, mental health, behaviors, and victimization.

The COVID-19 pandemic has resulted in substantial environmental and social changes in everyone’s life, including adolescents. If the extensive negative impact of the COVID-19 pandemic and its restrictions on adolescents’ well-being, behaviors, and health are recognized, then suitable social and healthcare strategies could be initiated to meet the needs of adolescents. Adequate support and help for adolescents could have a positive impact on our future.

A rapid literature review of research on children’s and adolescents’ mental health during the COVID-19 pandemic [6] described results drawn from six studies, for which the data collection was made during February and March 2020. Their findings indicated that symptoms of depression and anxiety increased in youth due to the COVID-19 pandemic. Saggioro de Figueiredo and colleagues [7] presented a discussion paper on the negative effects of the COVID-19 pandemic on children’s and adolescents’ mental health. They highlighted the biological, environmental, and social factors that simultaneously influence brain development, and in this way, children’s and adolescents’ behavior and well-being. The interconnecting factors of stress with neuroinflammation, social isolation with diet, social behavior, social inequalities with psychological distress, and more, are inevitably undergoing changes during the COVID-19 pandemic and will affect child and adolescent well-being [7]. Changes in psychosocial functioning during the COVID-19 pandemic in (Swedish) adolescents were investigated by Kapetanovic and colleagues [8]. Their data were collected during the summer holiday of 2020, after only a few months in the pandemic, from a country where restrictions were scarce and mild compared to other countries that had gone into total lockdown. They found that most adolescents did not report changes in their psychosocial functioning and in their risk behaviors, while indicating a lower frequency of victimization but still poorer mental health during the COVID-19 outbreak. A systematic review summarized the results of 16 studies (conducted between February 2020 and May 2020) on the impact of COVID-19 on adolescents’ mental health [9]. More than 40% of the reviewed studies originated from China, and the rest originated from industrialized countries. While the review highlighted the importance and protective effects of positive coping strategies and family support, the overall conclusion warned about the potential negative impact of the pandemic on adolescent mental health [9]. Unfortunately, the pandemic and its serious restrictions pervaded the whole year of 2020 and even the beginning of 2021, when mass vaccinations began globally.

The question has been raised: what are the effects of long-term COVID-19 restrictions on adolescents’ mental health, risk behaviors, psychosocial functioning, and victimization? We suggest that adolescents’ response to the long-term (9–11 months) restrictions as a consequence of the COVID-19 pandemic would be unique to their generation and culture; therefore, it is important for it to be described from a multicultural perspective.

Focusing on five countries (Vietnam, Serbia, Sweden, Morocco, USA) across four continents, each representing a specific cultural and socioeconomical sample in our study population, our aim was to overcome the generic problem that most social and behavioral studies analyze data from a narrow sample [10]. Based on a meta-analysis of over 700,000 college students’ responses to the COVID-19 pandemic [11], we hypothesized that adolescents’ responses to how much COVID-19 impacted their lives will differ in participating countries; for instance, adolescents in low-income, developing countries are impacted more by the COVID-19 restrictions (e.g., lockdown and isolation). We also hypothesized that the majority of adolescents from our multinational sample would report increased stress, attention problems, and higher levels of depression and anxiety, and that female students would be affected more strongly. Considering changes in risk behaviors, substance use, and norm-breaking actions, previous studies had conflicting results (with some reporting a decrease in them [8] and others indicating an alarming increase [12] during the first half year of the pandemic). We hypothesized a general increase in risk behaviors and a general decrease in victimization.

The goal of the present study was to describe the self-rated impact of the COVID-19 pandemic’s restrictions on adolescents’ behaviors, mental health, social functioning, and victimization from a global perspective and to identify gender differences.

## 2. Materials and Methods

### 2.1. Study Design and Procedure

This study had a cross-sectional design, selecting study participants by a non-probabilistic method and collecting responses on structured self-report instruments. The Mental and Somatic Health without borders (MeSHe) project [13] is an international study based in Sweden and approved in each participating country by their ethical board. The completion of the MeSHe electronic survey (available in relevant languages) was voluntary and anonymous. The survey was open from September 2020 to February 2021 in each of the five participating countries.

The participating countries’ economy, culture, and COVID-19-related restrictions differed significantly. Sweden: a highly developed Scandinavian country where restrictions were considered mild and few compared to those in other countries; Serbia: a quickly developing, upper-middle income Southeastern European country where restrictions were timely and serious from mid-March to mid-May 2020 with a complete lockdown; Vietnam: a lower-middle income nation with one of Southeast Asia’s fastest-growing economies, where COVID-19 restrictions came very early and included a total lockdown of society that lasted two weeks from the beginning of April 2020; Morocco: a developing North African country, where the COVID-19 pandemic was met with a total lockdown of society from mid-March to June 2020. This total lockdown was accompanied, for the first time in the country’s history, with material compensation for people without work and those who lost their jobs due to the pandemic; and finally, the USA, with most responses (40.3%) from New York State, where restrictions started earlier, in March 2020 and were longer and stricter than in other states. Until April 2021, there were similar restrictions in each country, such as limited social gatherings, face mask wearing in indoor public spaces, partial online and partial small-number in-person attendance in elementary and/or high schools, and public institutions, restaurants, and other businesses working limited hours with a limited number of people.

The MeSHe survey consists of validated questionnaires in which young people rate their own mental and physical health, aggressive, antisocial, and self-harm behaviors, personality, intensity, and frequency of leisure time physical activity, and mood. They also answer questions about the extent to which the COVID-19 pandemic has changed their behavior, mood, psychosocial functioning, and victimization [9]. In the present study, we report only the results of the responses to the COVID-19-related questions.

At the end of the MeSHe online survey, we included information about resources and links to support organizations available to everyone in the community (this list of organizations was adapted to each country).

### 2.2. Study Population

Contact with high school students was made via their schools (in Vietnam and Serbia, part of the data from Morocco, and a minor part of the data from Sweden and the USA) and via Facebook and Instagram (in Sweden, USA, and part of the data in Morocco). The national samples were as follows: the samples were representative for Sweden; in Serbia, participants from 15 cities and their surroundings were included; in Vietnam, the samples were mainly from Hanoi and surrounding cities; in Morocco, they were mainly from Tetouan and surrounding cities; and in the USA, they were from New York State, Texas, New Jersey, Connecticut, Florida, California, and Washington DC.

According to the original plans, we would have collected data in/through high schools in designated cities, ensuring representativeness for that city’s high school student population. However, during autumn 2020, we had to recognize that high school teachers and heads of schools were overwhelmed and could not participate in research studies. Therefore, we switched to reaching as many students by social media as we could in those countries where the response rate was very low when using designated high schools (Sweden, Morocco, and the USA). While this strategy increased the number of responses, it also decreased the generalizability and representativeness of the sample.

Between September 2020 and February 2021, we collected 5341 complete responses from five countries. A total of 1644 from Sweden, 1608 from Vietnam, 1162 from Serbia, 589 from Morocco, and 338 responses from the USA. Responses from adolescents under or over the age span (15–19 years old) defined in the protocol approved by the ethical boards were not included. The final number of participants was 5114 (1534 from Vietnam, 1108 from Serbia, 1608 from Sweden, 541 from Morocco, and 323 from the USA), consisting of 15 to 19-year-old high school students (37.0% male, 61.8% female, and 1.2% non-binary) with a mean age of 16.69 (SD = 1.01). The mean age was not different between genders: males M = 16.77 (SD = 1.01), females M = 16.65 (SD = 1.02), and those of non-binary gender M = 16.48 (SD = 0.99).

### 2.3. Measures

COVID impact. One item measured the personal overall impact of COVID-19 on the adolescents’ lives, expressed on a numeric analogue scale, between 0 “slightly or no affect” and 10 “has affected me immensely”.Changes in Adolescents’ Behaviors. Adolescents reported changes concerning substance use (4 items), relations with family and friends (6 items), everyday life situations (5 items), and norm-breaking behaviors (2 items). Responses to each item were measured on a 5-point Likert scale ranging from 1—decreased a lot, 2—decreased somewhat, 3—about the same as before, 4—increased somewhat, and 5—increased a lot. The participants could choose the option “I did not do this before the outbreak and have not started” and this response was coded as 0.Changes in Adolescents’ Mental Health. Ten items from the “Experiences Related to COVID-19 instrument” [14] were used to assess adolescents’ reported changes in sleep, stress, satisfaction, loneliness, involvement in society, and a different affect. The internal consistency was acceptable for this scale in the Swedish adolescent population (α = 0.82) [8], while it was low in our global study population (α = 0.53). The items were measured on a 4-point Likert scale ranging from 1 (do not agree at all) to 4 (agree completely).Changes in Adolescents’ Victimization. Changes in the frequency of victimization were assessed with five items selected from the Swedish Crime Survey [15] and previously used in an epidemiological study by Kapetanovic et al. [8]. The items assessed physical violence, threats, and sexual harassment (3 items) and online victimization (2 items), measured on a 5-point scale ranging from 1 (decreased a lot) to 5 (increased a lot). The scales previously showed acceptable internal consistency (α = 0.92) [8], which was similar to ours (α = 0.92).

### 2.4. Data Analysis

Since the measure “Changes in Adolescents’ Behaviors” was not previously tested for its psychometric features, it was first tested for its structure and internal consistency reliability by Cronbach’s α. A principal component analysis (PCA) with oblimin rotation was applied to test how 17 items behaved together and to see which items belonged to separate components. In a series of repeated PCAs, a three-component solution explained a total of 51.3% of the variance, with component 1 contributing 21.9%, component 2 contributing 18.1%, and component 3 contributing 11.2%.
Factor 1 related to risk behaviors included the items consuming alcohol; getting intoxicated by alcohol; smoking cigarettes; staying outside/being in the city without your parents’ knowledge; being outside and (for example) taking walks; with Cronbach’s α of 0.66.Factor 2 salutogenic approaches included the items having the opportunity to be in control over my daily life; keeping up with school projects and/or work; spending time doing things that I did not have time to do before; working out or exercising; spending time with family; and taking part in fun activities; with Cronbach’s α of 0.66.Factor 3 related to norm-breaking included the items stealing from shops, people or from your own or someone else’s home; and harassing someone on the internet using written language or uploaded pictures and/or videos; with Cronbach’s α of 0.64. The overall reliability for the remaining items (illicit drug use including prescription drugs used for reasons other than prescribed; staying in contact with relatives and friends over the phone/internet; staying connected with friends through social media or video games; arguing/fighting with my parent (or) parents; meeting up with friends in real life) was α = 0.60.

Respondents reported the extent to which their behaviors have changed, comparing them before and after the COVID-19 outbreak. Based on these responses, the percentage of change (“decreased a lot”, “decreased somewhat”, “decreased”, “increased”, “increased somewhat” and “increased a lot”) or no change (“about the same as before”) for each behavior was calculated. A chi-square test was used to compare the prevalence of changes in adolescents’ behaviors, psychosocial functioning, and victimization between male and female genders. Unstandardized odds ratios (OR) were used to quantify the association between the prevalence of these variables and gender.

IBM^®^ SPSS version 27 statistical program was used to analyze the data (IBM, Armonk, NY, USA).

## 3. Results

### 3.1. The Overall Impact of COVID-19 on Adolescents

The median score of the overall impact level of COVID-19 in the global adolescent population was five (on a VAS scale between zero and 10).

In total, 1778 males, 3023 females, and 55 non-binary gender participants responded to this question, reaching a median score of four in males (M = 4.19, SD = 2.96) and females (M = 4.61, SD = 2.90) and a median score of five in non-binary (M = 5.60, SD = 3.28) adolescents.

Figure 1 shows the proportion of young people by gender grading the impact of COVID-19 on their everyday life. Significantly (*p* < 0.001) more male than female or non-binary gender students reported no or very low impact of COVID-19 on their life. More females than males estimated a higher impact (5, 6, 7, 9, and 10 on a VAS scale) of the COVID-19 outbreak on their life (*p* = 0.002) and a large proportion (*p* < 0.001) of non-binary gender students reported that COVID-19 affected them a lot or immensely (7 and 10 on a VAS scale).

The self-reported impact of COVID-19 restrictions on young people’s everyday lives has differed depending on the nationality of the respondents (Figure 2). The highest impact (median at eight) was reported from students from the USA; importantly, this sample is not representative of the USA and is the smallest in our study. In the other four countries (Sweden, Morocco, Serbia, Vietnam), the median was five or below. With a comparable number of respondents, we had responses from Sweden, Serbia, and Vietnam. Among these three countries, Swedish adolescents reported the lowest impact on their everyday life by COVID-19 (Figure 2).

### 3.2. Changes in Adolescents’ Behaviors

We explored changes in adolescents’ behavior compared to their behavior before the COVID-19 outbreak. Table 1 summarizes the proportion in the multinational sample (N = 5114) of adolescents reporting any changes.

The majority (over 70%) of adolescents reported that the time they spent outside and how much they met with friends in real life has changed. Over 50% of them reported some kind of change in their behaviors in terms of spending time with their family, staying out without their parents’ knowledge, spending time with their friends and doing things they had not had time to do before, keeping up with schoolwork, having the opportunity to control their daily life, and exercising.

When describing the changes by the quality in each item (except changes in smoking and spending time with family), there were significant differences between the proportion of adolescents who decreased and those who increased each behavior. Figure 3a–d illustrates the proportion of adolescents decreasing or increasing their behavior as a consequence of the COVID-19 pandemic, considering the 17 questions.

A significant proportion of students decreased their time being outside (41.7%) (Figure 3a) and meeting friends in real life (59.4%) (Figure 3d). At the same time, a substantial proportion of them increased their time doing things they had not had time for before (49.3%) (Figure 3b) and their time staying connected with friends through social media and video games (44.9%) (Figure 3d), and with relatives through phone or the internet (41.7%) (Figure 3d). Almost a third of the young people increased working out (32.6%) (Figure 3b) and had the feeling that they have the opportunity to control their everyday life (31.2%) (Figure 3b). A similar proportion increased spending quality time with their family (30.5%) (Figure 3b), while a fifth reported increased arguments and conflicts with their parents (23.6%) (Figure 3d).

A much smaller proportion of adolescents reported changes in substance use habits and in norm-breaking behaviors, as the majority of them reported that they had not had this type of behavior before or after the COVID-19 outbreak. However, about 10% of the adolescents reported a decreased use of alcohol (13.3%) (Figure 3a) and decreased number of alcohol intoxications (9.8%) (Figure 3a), while 3.3% of them reported an increased frequency of illicit prescription drug use (Figure 3d). Of those few who engaged in norm-breaking behaviors, 2.6% reported a decrease in stealing, while 1.7% reported an increase in harassing others on the internet (Figure 3c).

These changes were distributed differently between male and female students. The number of female adolescents responding to these items varied between 3071 and 3103, while the number of male students varied between 1813 and 1835, and between 58 and 59 of non-binary students. We do not report the proportion of non-binary gender students who reported any changes because their number was low (between 0 and 30) compared to the other genders.

The proportion of those decreasing their behavior of getting intoxicated by alcohol was higher in male students (OR = 1.31 CI = 1.01–1.55) compared to females. The proportion of those who decreased spending quality time with their family (OR = 1.26, CI = 1.03–1.24), staying outside (OR = 1.13, CI = 1.06–1.21), meeting with friends (OR = 1.18, CI = 1.13–1.25), and having the opportunity to control their everyday life (OR = 1.22, CI = 1.10–1.35) was significant in female students.

The proportion of those increasing illicit drug use (OR = 1.60, CI = 1.17–2.16), staying outside without their parents’ knowledge (OR = 1.28, CI = 1.06–1.55), staying in contact with relatives on the phone/internet (OR = 1.28, CI = 1.06–1.55), meeting up with friends in real life (OR = 1.30, CI = 1.12–1.50), staying connected with friends through social media or video games (OR = 1.13, CI = 1.06–1.20), and harassing someone on the internet (OR = 1.65, CI = 1.08–2.52) was significantly higher in male than in female adolescents. However, the proportion of females was significantly higher than males in the increase of spending quality time (OR=1.13, CI=1.03–1.23) or arguing with their family (OR = 1.34, CI = 1.20–1.50), spending time with things they had not had the time to before (OR = 1.23, CI = 1.16–1.31) and keeping up with schoolwork (OR = 1.16, CI = 1.05–1.28).

### 3.3. Changes in Adolescents’ Mental Health

Participants rated the changes in different aspects of their mental health by comparing them in the present and before the COVID-19 outbreak (Figure 4).

About two-thirds of the high school students did not feel angrier (60.9%), sadder (61.5%), or lonely (60.5%), and more than half did not feel more anxious (55.0%), did not get into arguments more often (59.5%), did not mark changes in their sleep patterns (57.1%–58.4%), and even felt more involved in society (55.7%) after the COVID-19 outbreak. However, about half of them also felt more stressed (52.3%) and less content or fulfilled (50.1%).

A significantly larger proportion of female students reported feeling more anxious (*p* < 0.001; OR = 1.15, IC = 1.07–1.23), depressed or sad (*p* = 0.001; OR = 1.14, CI = 1.06–1.23), and sleeping more irregularly (*p* < 0.001; OR = 1.17, CI = 1.09–1.26) compared to male adolescents, while a significantly (*p* < 0.001) larger proportion of males reported that they slept as well as they did before the COVID-19 outbreak.

### 3.4. Changes in Adolescents’ Victimization

About 10% of the study population of adolescents reported that they have been victims of some kind of criminal offense during the past year. Figure 5 illustrates the proportion of students reporting that they have been victims of physical assault (10.3%), threat (11.5%), sexual assault (12.2%), cyber bullying (12.5%), or defamation (8.9%).

Physical assault and defamation were reported by a significantly larger proportion of male than female adolescents (*p* < 0.001; OR = 1.83, CI = 1.55–2.16 and *p* = 0.004, OR = 1.31, CI = 1.09–1.56, respectively). On the other hand, a significantly larger proportion of female than male students reported that someone had groped or touched them in a sexual manner without their consent (*p* < 0.001; OR = 1.77, CI = 1.48–2.11).

To measure if and how the COVID-19 restrictions impacted the frequency of the different types of offenses against adolescents, they were asked to report if these types of occurrences had been happening to them more often since the COVID-19 outbreak.

Around 15% of the adolescents reported a change in the frequency of the different types of victimizations. The proportion of adolescents reporting a decrease in the occurrence of physical (14.3%) or sexual assault (14.7%), online abuse (13.1%), threats (13.5), and defamation (13.3%) since the outbreak of COVID-19 was significantly more than those reporting an increase in these events (Figure 6). Female students were likelier to report an increased occurrence of physical (*p* = 0.036; OR = 1.49, CI = 1.02–2.17) and sexual (*p* = 0.013; OR = 1.49, CI = 1.08–2.04) assaults and threats (*p* = 0.002; OR = 1.56, CI = 1.17–2.07) in their lives since the COVID-19 outbreak. A higher proportion of students with a non-binary gender had reported changes generally and an increase of being victimized; however, the few numbers of non-binary gender students were not representative for statistical analyses.

## 4. Discussion

### 4.1. COVID-19 Restrictions’ Impact on Adolescent Mental Health, Behaviors, and Psychological Functioning

The MeSHe project provided an opportunity to assess the impact of the COVID-19 pandemic on adolescents’ experiences in different countries around the world. Participating countries differed substantially not only in their developmental stage, political, economic, social systems, and culture, but even in the type, rigor, and duration of restrictions in response to the COVID-19 pandemic. The study sample included high school students from two highly developed countries (38% of the total study population), one with quick and strict restrictions (USA), and one with milder and somewhat belated actions (Sweden). The rest of the study population consisted of high school students from three developing countries. Responses from Serbia (22%) counted as an upper-middle income European, those from Vietnam (30%) as a lower-middle income Asian, and from Morocco (11%) as a developing Arab-African cultures’ and environments’ moderating effects on our results. This multinational/multicultural study sample made the present study a unique and valuable addition to the growing list of psychosocial and behavioral studies on the subject.

There was one independent question in our survey that asked the respondent to estimate the overall impact of the COVID-19 pandemic on his/her life on a visual analogue scale, where zero represented slight or no effect and 10 represented an immense effect on the respondent’s life. When comparing the responses to this question from different countries, the well-known and expected cultural and socioeconomic influences were detectable. As expected, Swedish high school students (mild restrictions in a highly developed, democratic country) reported the lowest impact of COVID-19 restrictions on their lives. Students from developing countries reported a higher impact. These results are in accordance with previous studies and predictions, which have warned about an increased gap between high, middle, and low-income countries affecting the mental health of children, adolescents, and young adults after the COVID-19 pandemic [16]. Our results suggest that socioeconomic resources and the level of restrictions have indeed modulated the impact of COVID-19 on adolescents’ lives. However, we emphasize (as we discuss in detail later) that our data are not suitable for reaching generalized conclusions. Interestingly, going against all expected results, those few students responding to our survey from the USA (mainly New York State) reported a significantly higher impact of COVID-19 restrictions on their lives than any other participating country’s respondents. While again emphasizing that the USA sample was extremely small in our study and not representative, we are obliged to discuss this result. The USA is a highly developed country. We know that those students who responded to our survey had access to social media, stable internet, and parental consent to participate. Considering resources and possibilities to participate in our survey, the low number of respondents from the USA was already a sign of a distinct pattern of response to COVID-19 restrictions. Przybylski, Lewis, Ijzerman, and colleagues [17] summarized the pitfalls today’s human-science studies often fall into, with one being the fact that studies should account for deeper cultural, historical, political, and structural factors that have relevant moderating effects. In accordance with this, our result suggests that besides the socioeconomic resources of a country, cultural, political, and structural protective and risk factors should be seriously considered in future studies focusing on the impact of COVID-19 restrictions. However, results from the present dataset cannot be generalized.

Because of the uneven sample sizes, the non-representative nature of the samples from the participating countries, and the fact that the COVID-related inventory’s structural and cross-cultural validity and reliability have not been previously tested, we did not aim to present comparative analyses between countries regarding COVID-19-related changes in adolescents’ behaviors, mental health, and victimization. We admit that it is vitally important to compare adolescents’ responses to the COVID-19 restrictions between these nations; however, there is also a significance to having a comprehensive picture and reporting a multinational/multicultural impact. Therefore, in the present study, we did not focus on national or cultural differences, or differences in the length or extent of the restrictions; instead, we wished to give an overall picture about the impact of the past year on adolescents’ lives.

Healthcare professionals have predicted, and the first published studies have shown, that the COVID-19 pandemic would cause a substantial negative impact on adolescents’ lives in the form of an increased prevalence of post-traumatic stress disorder (PTSD), depression, anxiety, grief-related symptoms, and sleep problems [18,19,20,21,22,23].

In our study, about two-thirds of adolescents did not report changes in their mental health related to feelings of increased anger, sadness/depression, or loneliness. More than half did not feel more anxious and did not experience any change in their sleep patterns. However, about half of the participants felt more stressed. In contrast to previously published results [9] and our hypothesis, our findings suggest that the majority of high school students did not experience any change in their usual emotional state and sleep pattern, despite increased general stress. When changes in mental health were reported, female students had a significantly higher risk of reporting increased anxiety, feeling blue or sad, and sleep disturbance. To obtain a more complete picture of female adolescents’ wellbeing, we need to remember that females were likelier to report a decreased opportunity to control their everyday life and stay outside and meet with friends. They also reported more arguments with their family, spending more time with things they have not had the time to before, and keeping up with schoolwork.

During the summer of 2020, older school-age adolescents (15–18 years old) reported increased stress caused by physical distancing during the summer and the anticipated challenge of returning to school during the fall [23]. The increased stress level seems to be a constant result independent of the length of the restrictions. In our sample, in agreement with what was reported in Swedish [8], Canadian [23], and Swiss [24] samples, and in an international meta-analysis on college students [11], a significantly larger proportion of female students reported feeling more anxious, depressed, or sad, and sleeping more irregularly after the COVID breakout, compared to male adolescents. In our study population, there were almost twice as many female as male respondents, while other researchers worked with a more balanced gender distribution in their samples [8,23], and still reported worse mental health in females. Therefore, we believe that the gender-specific results did not emerge from the uneven gender representation in our sample. Instead, the higher levels of stress, psychological distress, and mental health complaints in female adolescents compared to male adolescents seems to be a consistent finding in different cultures [25,26,27,28,29]. It is hypothesized that a higher level of anxiety and perceived stress in female adolescents originates from the higher demands placed on them in different arenas of life, such as family and school [30,31]. This hypothesis fits our results well: during the COVID-19 pandemic, female adolescents were at higher risk of experiencing increased stress, mood changes, and sleeping problems, and at the same time, they had increased focus on their school performance and time spent with their families, while still feeling less control over their lives.

While Saggioro de Figueiredo et al. [7] suggested that long-term social isolation and changes in daily routines combined with other stressful life events can have both short-and long-term consequences on adolescents’ lives, our study assessing changes in adolescents’ lives after a 9 to 11-month-long restriction period showed that more than half of them felt even more involved with society, spent more quality time with their family, and did not get into arguments with their family more often. At the same time, a substantial proportion of them increased their time doing things they did not have time for before, stayed connected with friends through social media and video games, and communicated with relatives through phone or internet. Even if our study population included a variety of nations, it seems that the majority of adolescents could adapt to the isolation caused by the restrictions. The total lockdown and closed school campuses did not result in social isolation in today’s generation of adolescents, as their social media and online contacts were already substantial before the pandemic. It is likely that adaptation to these changes was easier for them compared to older generations, who had no equipment or knowledge of methods for staying in touch with friends and family using digital resources [32].

We have to acknowledge again that our results cannot be generalized to a global trend; instead, based on a multinational sample, we aim to give an overview and some specific examples of how adolescents have responded to the long-term restrictions during 2020. Behind this overall picture, there are, as previously mentioned, cultural and economic differences, where low-income developing countries’ adolescents were affected more seriously by social isolation without possibilities for online education and keeping in contact with others using social media and the internet, combined with less access to healthcare and social services [2]. In addition, it is important to recognize that even though the majority of the adolescents did not change their daily habits, and the COVID restrictions did not affect them negatively, there is still a substantial proportion of adolescents who indicated negative changes in their mental health, psychosocial functioning, and risk behavior.

In our sample, over 40% of students reported that they had decreased their time outside, yet almost every third young person indicated that they had worked out more and felt that they had the opportunity to better control their everyday life. Once again, there is a possible explanation that today’s youth understand the importance of physical activity in maintaining physical as well as mental health, and parents encourage children and adolescents to exercise even when ordered to stay at home [33]. Some studies have raised concerns about the possibility that social isolation and stay-at-home orders issued in cities across the globe would reduce the opportunities for physical activity among youth and would lead to weight gain and increased frequency of obesity [34]. However, according to our results, the majority (over 70%) of adolescents increased or did not change the time they exercised during the COVID-19 restrictions, which may have contributed to the finding that the majority of them did not report worsening mood, anxiety, or sleep patterns. Studies at the beginning of the pandemic warned that school closures and home confinement could lead to decreased physical activity and worsen psychological well-being in children and adolescents [35], and others reported such changes in a small study population (N = 125 university students in Canada) [36]. A review [37] examined physical activity and psychological distress in adolescents during the first half of the year 2020. They included four articles (three from China and one from Spain) on physical activity and mental health during COVID-19. Three of the studies showed a decrease in physical activity because of COVID-19, and two of them [38,39] stated that mental health deteriorated due to decreased physical activity. As we presented earlier, in our study we did not see decreased physical activity or worsening mental health in most countries. We can only speculate as to the reason why. It may be that until autumn 2020, when a new academic year started with fully online or mixed online and campus education, schools could adapt and prepare their curriculum to maintain and support high school students’ physical activity. It is also possible that the majority of adolescents in our study population already incorporated exercise into their everyday lives, and exercise was a protective factor against worsening mental health during the long-term COVID-19 restrictions.

Another prediction that was not supported by our results is the expectation that adolescents would increase their substance use as a coping strategy during extremely stressful or traumatic environmental changes in their lives [7,18]. A few studies examining adolescents’ substance use during disasters [12,40] suggested that extreme life events (including the COVID-19 pandemic) were associated with an increase in illegal substance use and abuse and an increased risk for the development of addictions. Considering the above findings and predictions, our results related to changes in substance use were somewhat unexpected. Based on our findings, we have to reject our hypothesis suggesting increased substance use and risk behaviors in adolescents as a consequence of long-term restrictions. The majority of adolescents (76–94%) did not change their substance use behavior during the pandemic, and for the absolute majority, it meant that they had not used substances before the pandemic or during the pandemic. Although about one-fourth (23.5%) of the students changed their alcohol consumption habits, more than half of these adolescents actually decreased their alcohol use. It is well known that increased stress, fear, imbalance between different components of everyday life, social isolation, and traumatic events alone and together can increase substance use [41,42,43,44]. Why did we not find a higher proportion of adolescents seeking relief with alcohol use? It could be that self-reports suffer from the bias of those reporting not wanting or daring to report facts that may collide with social and moral expectations. However, other studies and governmental reports describing adolescents´ alcohol and drug use were also based on (fully or at least partially) self-reports. Our data collection offered total anonymity for the participants; therefore, we think that false reporting is negligible. One important difference between previous studies and ours is that during the COVID-19 pandemic, adolescents’ social lives moved predominantly to the internet and social media. One possible explanation is that adolescents’ opportunities to go out and meet friends, to party, or to spend the day or evening in the absence of adult control were significantly reduced. Our results support this fact, as a large proportion of young people reported a decreased number of events of going out and being with friends in real life, while reporting increased time (quality time) with their parents. Even if we do not question all those stressful life events from the COVID-19 pandemic having an impact on adolescents’ lives, it seems the majority of them could not/did not increase alcohol use as a coping strategy. Another possible explanation could be that adolescents’ addictive behavior shifted from substance use to smartphone and internet addiction [9]; however, we have not assessed these behaviors in our survey.

The use of prescription drugs, however, showed a somewhat more alarming tendency. More than 60% of those 5% of adolescents who reported a change in their prescription drug use reported increased use. This is an important result on which we should reflect. The exponentially increasing number of young people using and being addicted to opioid analgesics or stimulants was a highly relevant worldwide problem before the COVID-19 pandemic [45]. The world has been observing an opioid epidemic and a new wave of the amphetamine epidemic, mostly in the USA but also worldwide [46,47,48,49]. In a recent study on adults, the convergence of the opioid epidemic with the COVID-19 pandemic was found. This study showed that there was an increased use of a dangerous combination of high-risk drugs [50]. We suggest that when societies start functioning as before and high school students return to their usual campus learning and outdoor life, there may be an increased need for school, healthcare, public health, and social resources dedicated to supporting adolescents who developed prescription drug abuse during the pandemic and who will be in need of treatment and support.

### 4.2. Victimization

According to a meta-analysis, the estimated global prevalence of self-reported physical and emotional victimization was 16% and 18%, respectively [51]. It has also been shown that it is more common to be victimized in adolescence as compared to childhood and more common to be victimized more than once than to be victimized on a single occasion [52]. In our study, about 10% of the study population of adolescents reported that they had been victims of some kind of criminal offense during the past year, which would indicate a general decrease in victimization during the COVID-19 restrictions. Often, victimization starts with personal contacts in school or peer contexts among adolescents and continues online [53]. Following this context, it was predicted and shown early in the pandemic that the risk of being subjected to victimization had decreased [8]. However, others have focused on the fact that with the increased use of social media during the lockdown, increased numbers of adolescents have been victims of cyberbullying, including “sexting”, sexual comments on young girls’ pictures, and insulting each other [54]. In our study, around 15% of the adolescents reported that the frequency of victimization had changed since the COVID-19 outbreak, and the great majority of them indicated a decreased frequency of being victimized. This result may support the hypothesis that a dramatic reduction in physical/school contact with peers and staying at home leads to decreased victimization. In addition, the reported “no change” or “decrease” in the use of alcohol may be related to the decreased frequency of victimization of the adolescents. Surprisingly, according to the adolescents’ self-reports, even cyberbullying decreased for most of them.

Previous studies consistently report that more girls than boys are sexually victimized [52,55,56,57,58,59]. Gender differences are less clear and consistent when it comes to physical or verbal victimization, neglect, witnessing of violence, and property crime [52,60,61]. In our study, we could also recognize the typical gender differences in victimization, where male adolescents reported being the victim of physical assault significantly more frequently than females, while female adolescents reported being the victim of groping more often than males.

## 5. Strengths and Limitations

The present study is one of the first international attempts to describe changes in adolescent risk and norm-breaking behaviors, salutogenic approaches, mental health, psychosocial functioning, and victimization as a consequence of the COVID-19 pandemic and related long-term restrictions. The considerably large study population, the inclusion of several countries, and the timing of data collection are strengths of the present study. One of the most important limitations is the non-representative nature of the sample in most countries and the uneven response rate between the participating countries. Since the MeSHe survey included several well-known and validated instruments in addition to the COVID-19-related questions that are presented here, we plan to continue data analyses and present comparisons between the countries’ study populations in mental and somatic health, substance use habits, norm-breaking, and self-harm behaviors and investigate factors that may modify the associations, after controlling for cross-cultural validity of used inventories. We would like to emphasize again the fact that while we present results from an unusually diverse study population, that is not representative of the participating countries, with the exception of Sweden. Most of the data were collected from the main cities of the participating countries, and the majority of students had to have access to stable internet to answer a survey lasting about 35–45 min, possibly leading to a sampling bias of students with a higher socioeconomic status. Therefore, we did not report results by participating countries, except for one question in the survey, considering the impact of COVID-19 on adolescents’ lives. These results are not generalizable for the participating countries; however, they give an overview and specific examples of problem areas to focus on.

Being able to capture students who identify as non-binary is an important achievement; however, a limitation is that no meaningful comparisons could be carried out with this group of youth due to their low prevalence in the sample. Speaking about the gender distribution of our study population, it has to be noted that there were more female than male students responding to the survey. The statistical analyses used ensured that the uneven ratio of the genders did not affect the results.

While the COVID-19-related questions are not validated instruments to detect or diagnose clinically relevant mental illness, and their factors had a low internal consistency (Cronbach’s α under 0.7), the results reflect the adolescents’ symptoms and own experiences, providing a relative measure of their mental health. As the questions were formulated to compare their feelings, sleep patterns, and stress levels at present vs. before the COVID-19 outbreak, our results report the relative quality of these mental health items and the perceived effect of COVID-19 on them, not the absolute prevalence of these problems.

## 6. Conclusions

The majority of the high school students surveyed did not report changes in risk behaviors or worsening mental health symptoms. A predominant proportion of the adolescents increased their exercise, spent more quality time with their family, focused on things they did not have time for before, maintained social contacts with friends and relatives using the internet, and even felt more involved with their community after the COVID-19 outbreak. All types of victimization decreased during the COVID-19 pandemic.

Female adolescents and non-binary youth were impacted more negatively by the COVID-19 pandemic. A small, but definitely not negligible proportion of the adolescents increased their prescription drug use.

Our overall results suggest that in general, the majority of adolescents had several social and cultural resilience factors in their lives and found positive coping strategies to handle the extreme stress of the COVID-19 pandemic. From an overall perspective, we can conclude that the fears and warnings about the serious negative impact of the strict COVID-19 restrictions on mental health, risk behaviors, and victimization were not present in the majority of high school students’ lives in our multinational sample. While stating this, we would like to emphasize that the past year indeed negatively affected a small portion of adolescents, resulting in increased risk behaviors (substance use, norm-breaking behaviors), worsened mental health (sleep problems, amplified stress, anxiety, depression), and even increased frequency of being a victim of criminal offenses.

Future analyses should focus on comparing validated outcome measures in national results to assess changes in mental health and risk and protective factors. Improving scientific works’ evidence readiness levels [17] could make findings appropriate for guiding healthcare institutions and municipalities, schools, and social services in finding and meeting the needs of those young people who are affected negatively by the prolonged COVID-19 restrictions, and for developing adequate support and help for young people to ensure their healthy development for the benefit of our society.

## Figures and Tables

**Figure 1 ijerph-18-08755-f001:**
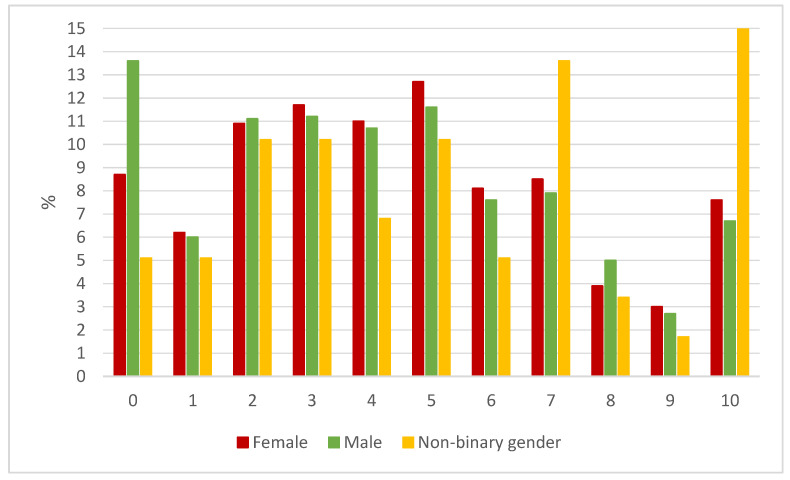
The overall impact of the COVID-19 outbreak on adolescents. Adolescents rated the overall impact of COVID-19 on their life on a numeric analogue scale, between 0 “slightly or no affect” and 10 “has affected me immensely”.

**Figure 2 ijerph-18-08755-f002:**
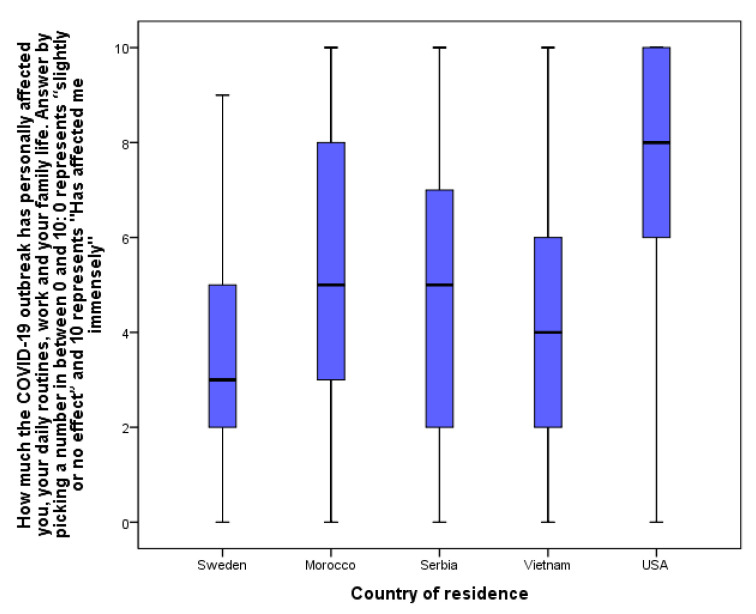
The impact of the COVID-19 outbreak on adolescents according to their country of residence. Number of respondents: N_Sweden_ = 1584; N_Morocco_ = 441; N_Serbia_ = 1079; N_Vietnamn_ = 1452; N_USA_ = 321.

**Figure 3 ijerph-18-08755-f003:**
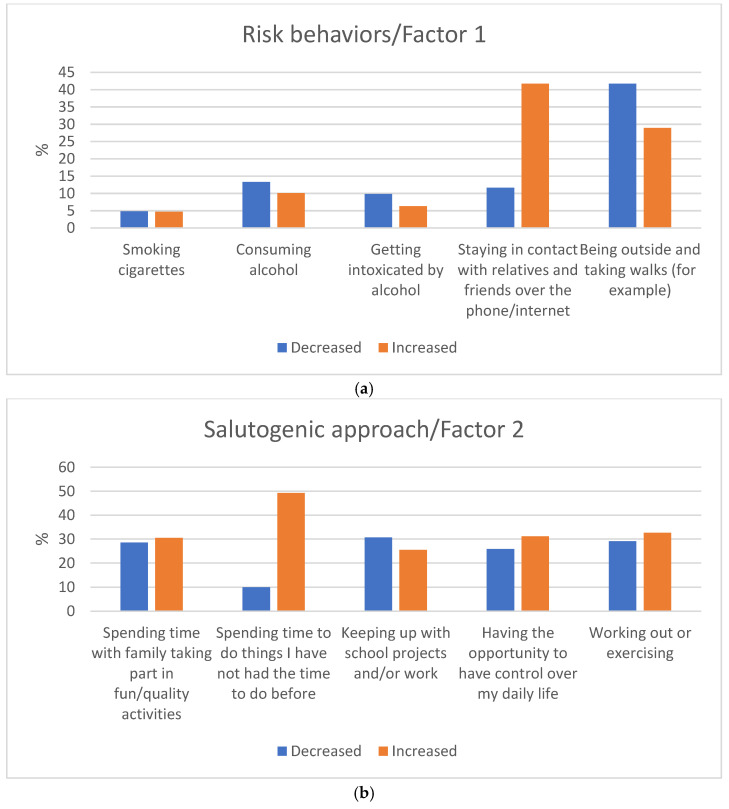
(**a**–**d**) Changes in adolescents’ risk behavior (Factor 1), Salutogenic approach (Factor 2), Norm-breaking (Factor 3), and Additional item in the multinational study population. (**a**) Changes in risk behavior of adolescents during the COVID-19 restrictions; (**b**) changes in salutogenic approaches of adolescents during the COVID-19 restrictions; (**c**) changes in norm-breaking behaviors of adolescents during the COVID-19 restrictions; (**d**) changes in additional behaviors of adolescents during the COVID-19 restrictions.

**Figure 4 ijerph-18-08755-f004:**
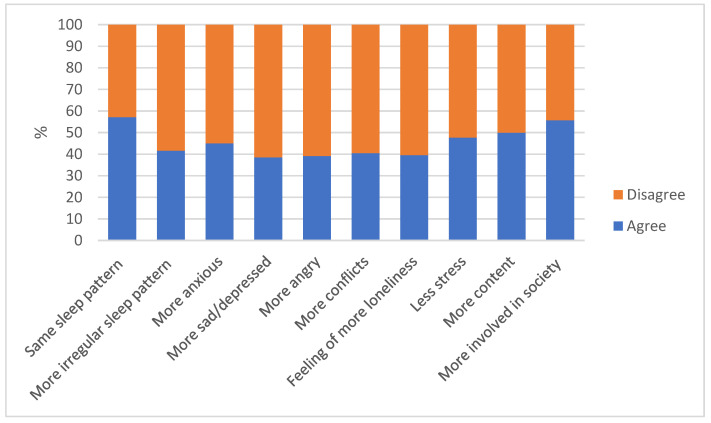
Changes in mental health factors caused by the COVID-19 outbreak in the global study population.

**Figure 5 ijerph-18-08755-f005:**
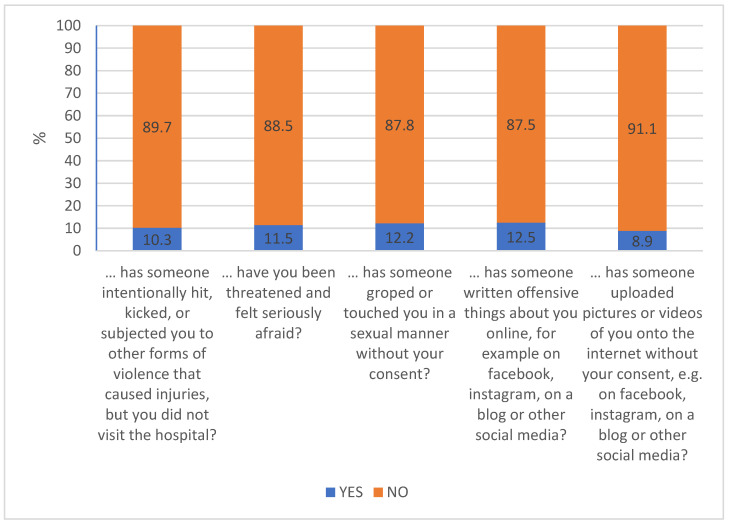
Presence of victimization during the past 12 months in the global study population.

**Figure 6 ijerph-18-08755-f006:**
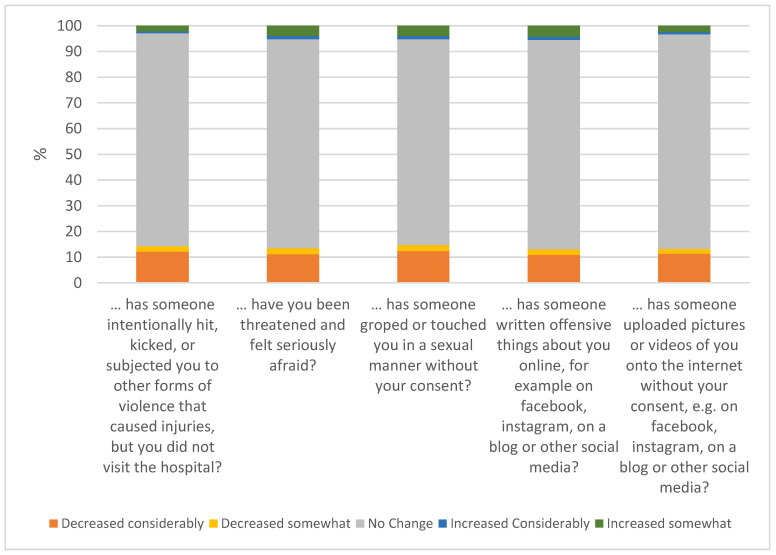
Changes in the frequency of victimization since COVID-19.

**Table 1 ijerph-18-08755-t001:** Responses about adolescents’ behavior change comparing frequencies before and after the COVID-19 outbreak.

	N between 5018–4962	No Change (%)	Change (%)
**Factor 1** **Risk Behavior**	Smoking cigarettes	90.5	9.5
	Consuming alcohol	76.5	23.5
	Getting intoxicated by alcohol	83.9	16.1
	Staying outside/being in the city without your parents’ knowledge	46.7	53.3
	Being outside and taking walks (for example)	29.4	70.6
**Factor 2** **Salutogenic Approach**	Spending time with family taking part in fun/quality activities	40.9	59.1
	Spending time to do things I have not had the time to do before	40.6	59.4
	Keeping up with school projects and/or work	43.8	56.2
	Having the opportunity to have control over my daily life	42.9	57.1
	Working out or exercising	38.3	61.7
**Factor 3** **Norm-breaking** **Behavior**	Stealing from shops, people or from your own or someone else’s home	95.7	4.3
	Harassing someone on the internet using text or uploaded pictures and/or videos	97.3	2.7
**Additional items**	Illicit drug use: including prescription drugs used for reasons other than prescribed	94.7	5.3
	Arguing/fighting with my parent (or) parents	66.7	33.3
	Meeting up with friends in real life	27.4	72.6
	Staying in contact with relatives and friends over the phone/internet	79.8	20.2
	Staying connected with friends through social media or video games	46.5	53.5

## Data Availability

Original data tables used for statistical analyses are available by contacting the principal investigator (NK).

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
