# Peer review of "Changes in Adolescents’ Psychosocial Functioning and Well-Being as a Consequence of Long-Term COVID-19 Restrictions"

_ijerph, 2021, doi:10.3390/ijerph18168755_

Round 1

Reviewer 1 Report

Dear Authors, 

Thank you very much for the opportunity to review this interesting work considering perceived changes of adolescents in their psychosocial functioning and well-being from 9 to 11 months after the COVID-19 restrictions. 

I think this work addresses important and innovative issues. In fact, trying to understand the effects of the pandemic on specific group populations is indeed a common challenge among researchers that interests the general population as well.

I would suggest the authors read the commentary (see below) to reflect on how effortful it is to transform behavioral research into policy and how caution we all must remain in the descriptions of our behavioral results. 

  • IJzerman H, Lewis NA Jr, Przybylski AK, Weinstein N, DeBruine L, Ritchie SJ, Vazire S, Forscher PS, Morey RD, Ivory JD, Anvari F. Use caution when applying behavioural science to policy. Nat Hum Behav. 2020 Nov;4(11):1092-1094. doi: 10.1038/s41562-020-00990-w. PMID: 33037396.

Despite the relevance of this work, there are major issues that should be addressed before the official publication in the International Journal of Environmental Research and Public Health. English writing should be checked in detail for misspellings and oversights.

Comments 

1. Introduction 

  • Line 103: The authors outline the main goal of the study without making specific nor general hypotheses about what they might expect. Authors should describe the hypotheses more in detail in this part in line with previous literature.

2. Materials and Methods

Study design and procedure

  • In the procedure it is not mentioned when the questionnaires are administered. The authors wrote about “long-term restrictions” . What does it mean precisely? Please provide this information which can be found later in the discussion part (line 420)
  • From lines 122 to 126: I think this information should be put in the “study population” paragraph.

Study population

  • How did the authors decide on the size of the sample? I would suggest conducting a power analysis before starting the study. If the authors did, please provide information about this in the paragraph on study population
  • The sample is unbalanced considering gender. Could it be that the results that the authors find for females are somehow influenced by their greater number (almost double) in the whole sample? Please discuss this point.

3. Data Analysis 

  • The validation of “changes in adolescents’ behaviors” is described in detail. However, the authors should provide a more detailed analytic plan considering the analysis conducted. Each step should be reported.

5. Discussion 

  • Line 377: same/similar phrase is reported in the introduction as well
  • Line 405: authors discuss gender differences, I think it would be appropriate also to discuss these findings in light of the unbalanced sample between females and males. Are other studies you mentioned unbalanced as well? In which ways?. Please provide clarification for this.
  • Line 462: The authors should provide explanations for the results not confirming previous studies. Please discuss this further.
  • Line 487: The authors state that “the majority of adolescents could not/did not increase alcohol use as a coping strategy”. Could you provide more explanation for this to clarify this aspect?
  • In general, I think that results should be reported in a more moderate way given the fallacy of the assessment through a self-report tool. The same goes for the conclusions that are drawn. 

Author Response

First of all, we thank the Reviewer for his/her time, encouraging words, and constructive criticism. We corrected spelling errors, and replied to all raised questions and concerns, hoping that it increased the quality of our study. We thank the Reviewer for mentioning the highly relevant and interesting publication that we were not aware of before. We have incorporated relevant aspects of that in our discussion. 

Comments 

  1. Introduction 
  • Line 103: The authors outline the main goal of the study without making specific nor general hypotheses about what they might expect. Authors should describe the hypotheses more in detail in this part in line with previous literature.

We acknowledge the absence of a clearly stated hypotheses. These were added to the current version (Lines 110-127)

  1. Materials and Methods

Study design and procedure

  • In the procedure it is not mentioned when the questionnaires are administered. The authors wrote about “long-term restrictions” . What does it mean precisely? Please provide this information which can be found later in the discussion part (line 420)

We updated the Procedure section as well, added the time period of collection, and moved the suggested information from the discussion to the description of the study design (Lines 133-134 and 140-157).

  • From lines 122 to 126: I think this information should be put in the “study population” paragraph.

We have moved this information (Line 169-184).

Study population

  • How did the authors decide on the size of the sample? I would suggest conducting a power analysis before starting the study. If the authors did, please provide information about this in the paragraph on study population

The current study was built on pilot data collected in Sweden. The MeSHe project is an ongoing project. According to the original protocol we would have collected data from students enrolled in several high schools in one designated city per country, ensuring representativeness for that city’s high school student population. However, during late Autumn 2020 we had to recognize that high school teachers and heads of schools were overwhelmed with the reopening and could not help us with the research project. Therefore, we switched recruitment to social media to reach as many students by as we could (in Sweden, Morocco, and US). While this strategy helped to get a high number of responses it also diminished generalizability and representativeness of the sample. We have now also described the sampling method in detail Lines 133-134 and 169-184.

  • The sample is unbalanced considering gender. Could it be that the results that the authors find for females are somehow influenced by their greater number (almost double) in the whole sample? Please discuss this point.

Thank you for pointing this out. We have discussed this limitation in the current version of our manuscript (Lines:495-499) and mentioned in the section of Strength and Limitations (Lines 671-674).

  1. Data Analysis 
  • The validation of “changes in adolescents’ behaviors” is described in detail. However, the authors should provide a more detailed analytic plan considering the analysis conducted. Each step should be reported.

Thank you for your comment. We added a paragraph in the data analysis section describing the steps taken to analyze the results related to the changes in adolescents’ behaviors.(Lines 246-250).

  1. Discussion 
  • Line 377: same/similar phrase is reported in the introduction as well

We have deleted this section from the discussion.

  • Line 405: authors discuss gender differences, I think it would be appropriate also to discuss these findings in light of the unbalanced sample between females and males. Are other studies you mentioned unbalanced as well? In which ways?. Please provide clarification for this.

We have discussed the uneven distribution of genders in our results both at discussion (Lines:495-499) and in the Limitation section (Lines 671-674).

  • Line 462: The authors should provide explanations for the results not confirming previous studies. Please discuss this further.

Thank you for encouraging us to reflect on our results and further discuss them. We have expanded the discussion and added a recent review. (Lines 554-566). 

  • Line 487: The authors state that “the majority of adolescents could not/did not increase alcohol use as a coping strategy”. Could you provide more explanation for this to clarify this aspect?

We have critically reflected on this result and discussed it adding on two other possible explanations. (Lines:584-589 and 598-600.)

  • In general, I think that results should be reported in a more moderate way given the fallacy of the assessment through a self-report tool. The same goes for the conclusions that are drawn. 

Thank you, we do agree, and changed our discussion and conclusions using expressions of “may, could”, and emphasized problems with generalizability several times, and rephrased the conclusion about using our results in policy making. 

Reviewer 2 Report

Changes in adolescents’ psychosocial functioning and wellbeing as a consequence of the long-term COVID-19 restrictions

Thankyou for giving me the possibility to read and review this paper.

This paper is very well written, and the arguments are very well developed. I think it should be published as is. It is a privilege to read a paper that requires no revisions. All I want is to ask the authors to think of adding a small sentence that explains ‘The final number of participants was 5114’ (lines 139/144).  The paper adds the data from the five countries and considers them as a whole data set.  While this is current practice, I am certain that the authors are aware that this eliminates focus from country complexity. I am suggesting to the authors that writing a few sentences arguing the benefits of the whole data set will make this paper complete.

Once, again thank you for giving me the possibility of reading this paper.

Author Response

Thank you for your time, kindness and encouraging words. Based on other reviewers' request, we added a country specific data on how much COVID-19 impacted adolescents on a VAS scale between 0 and 10.

We changed the introduction, results and discussion to reflect this.

Reviewer 3 Report

Comments for authors

Thank you for the opportunity to review “Changes in adolescents’ psychosocial functioning and wellbeing as a consequence of the long-term COVID-19 restrictions”. This manuscript examined the impact of COVID-19 on adolescents’ behaviors, relationships, mood, and victimization.This is an interesting and meaningful study. The study leveraged survey data collected from five countries (Sweden, the USA, Serbia, Morocco, and Vietnam) between September 2020 and February 2021. The study found that that the majority of adolescents could adapt to the dramatic environmental changes without being affected negatively. However the manuscript can be much improved.  Below, I discuss a few major concerns about the paper.

  1. The survey was conducted in five countries with very different culture and different pandemic restriction. The heterogeneity of data is expected. However it is shown how the heterogeneity was accounted in the analysis, for example, was mixed effect or clustering effect incorporated in the regression?

2.It appears that the response rates between male and female differ substantially (37.0% male, 61.8% female and 1.2% other sex).  Could the author explain the reason?

  1. In the data analysis, it is stated that “Odd ratios (OR) were used to quantify the association between the prevalence of these variables and gender”. However, what specific regression was not described. Also it is not clear if the OR is adjusted or unadjusted OR.

  1. In the discussion, the authors claimed “our results are highly generalized and aim to give an

overall picture on how adolescents have responded to the long-term restrictions during 2020, from a sample that includes a variety of international samples.” However, consider the sample size for each country, as well as the fact that the participants are mainly from big cities, the generalizability is really questionable.  I would suggest rephrase this statement.

  1. The study did not find many changes in substance use (The majority of adolescents (76%-94%) did not change their substance use behavior during the pandemic and for the absolute majority it meant that they had not used substances before the pandemic nor during the pandemic). The authors’ explanation is that the opportunities for adolescents to go out had been reduced during the pandemic. But self-report bias may also play a role here, since people are not reluctant to report negative outcomes.

  1. A large part of the discussion can go to the results section.

Author Response

Thank you for your comments and suggestions. We have adressed each in detail. 

  1. The survey was conducted in five countries with very different culture and different pandemic restriction. The heterogeneity of data is expected. However it is shown how the heterogeneity was accounted in the analysis, for example, was mixed effect or clustering effect incorporated in the regression?

 We did not perform regression analyses in our study, therefore no clustering effect incorporated to take this heterogeneity into account in the analysis. We used nonparametric test “Chi-square test” to compare the prevalence of changes in adolescents’ behaviors, psychosocial functioning, and victimization between male and female genders.

  1. It appears that the response rates between male and female differ substantially (37.0% male, 61.8% female and 1.2% other sex).  Could the author explain the reason?

Thank you for raising this interesting point. It has been shown that in general, women are more likely to participate in online surveys than men (Curtin et al 2000; Moore & Tarnai, 2002; Singer et al 2000). However, according to a recent study (Mulder, Joris, and Marika de Bruijne. 2019. “Willingness of Online Respondents to Participate in Alternative Modes of Data Collection.” Survey Practice 12 (1). https://doi.org/10.29115/SP-2019-0001.) based on almost 2000 respondent each over the age of 16, there was no significant difference in the willingness of women and men to respond to online surveys. We changed the discussion (Line:495-499) and the Limitation section (Lines 671-674) to reflect this.

3. In the data analysis, it is stated that “Odd ratios (OR) were used to quantify the association between the prevalence of these variables and gender”. However, what specific regression was not described. Also it is not clear if the OR is adjusted or unadjusted OR.

  As mentioned in the answer above, we used Chi-square test and no regression analysis was carried out. We reported unstandardized Odd ratios (OR), that is now clearly stated in Data Analyses section.

4. In the discussion, the authors claimed “our results are highly generalized and aim to give an overall picture on how adolescents have responded to the long-term restrictions during 2020, from a sample that includes a variety of international samples.” However, consider the sample size for each country, as well as the fact that the participants are mainly from big cities, the generalizability is really questionable.  I would suggest rephrase this statement.

Thank you for bringing this up. Indeed, our data does not fulfill the requirements of generalizability, and we now discussed this in detail. We have rephrased the sentence (Line 526)

5. The study did not find many changes in substance use (The majority of adolescents (76%-94%) did not change their substance use behavior during the pandemic and for the absolute majority it meant that they had not used substances before the pandemic nor during the pandemic). The authors’ explanation is that the opportunities for adolescents to go out had been reduced during the pandemic. But self-report bias may also play a role here, since people are not reluctant to report negative outcomes.

We have mentioned one possible explanation to this observation: the reduced opportunities to go out, party, and get access to alcohol. We do agree that self-reports are biased, because people avoid reporting facts which may collide with social and moral expectations. However, other studies and governmental reports describing adolescents´ alcohol and drug use are also based on self-reports. Or data collection offered total anonymity for the participants therefore we assume that false reporting is minimal.  We have now discussed this limitation (Lines: 584-689) and added on other possible explanation this in the discussion (Lines: 598-600).

6. A large part of the discussion can go to the results section.

We humbly ask the reviewer to let us keep our discussion comprehensive following the other four reviewers’ comments

The manuscript was language edited by an official language editing firm. 

Reviewer 4 Report

Review for Changes in adolescents’ psychosocial functioning and wellbeing as a consequence of the long-term COVID-19 restrictions

This paper presents adolescents’ reports on how the COVID-19 pandemic has changed their behaviors, relationships, mood, and victimization. Data collection was conducted in five countries (Sweden, the USA, Serbia, Morocco, and Vietnam). The sample (N=5114) comprised by high-school students (aged 15 to 19 years, 61.8% females) who had to respond to an electronic questionnaire. The results showed that students reported decreased time outside (41.7%), meeting friends in real life (59.4%), and school performance (30.7 %), while reporting increased time to do other things for which did not have time for before (49.3%) and using social media to remain connected (44.9%).

One third of the respondent reported that increased exercise and feeling that they have more control in their life, while a small proportion of adolescents reported substance use, norm-breaking behaviors, or being victimized.

Moreover, the overall COVID-19 impact on adolescent life was gender specific, that is there is a stronger negative impact on female students. The majority of adolescents were able to adapt the dramatic environmental changes. The finding of the present work will be beneficial for healthcare institutions, municipalities, schools, and social services, which care about the needs of those young people whose psychosocial functioning, stress or victimization experiences have been affected.

This work presents a survey with global interest. The paper is well written and the presentation is clear, providing some answers to some research question, which however are not clearly stated.

It would improve the paper if the research hypotheses or questions are posited in a separate section, so it would follow the formal style of scientific paper and help the reader to understand what is being sought.

Moreover, I would expect an extensive analysis of the data. The results are merely based on descriptive statistics, providing what is happening ‘at the average’.

I propose the authors to present some inferential statistics testing hypotheses on the relation among factors and some cross-cultural comparisons.

Author Response

This work presents a survey with global interest. The paper is well written and the presentation is clear, providing some answers to some research question, which however are not clearly stated.

Thank you for your time and review. We agree with this observation. We added our hypotheses to the introduction section.

It would improve the paper if the research hypotheses or questions are posited in a separate section, so it would follow the formal style of scientific paper and help the reader to understand what is being sought.

 We acknowledge the absence of clearly stated hypotheses. We added our hypotheses to the introduction section. (Lines 110-127). Presentation of the hypotheses also improved our discussion.

Moreover, I would expect an extensive analysis of the data. The results are merely based on descriptive statistics, providing what is happening ‘at the average’.I propose the authors to present some inferential statistics testing hypotheses on the relation among factors and some cross-cultural comparisons.

Indeed, we are aware of the need and curiosity about cross-cultural comparisons. Unfortunately, our datasets are not representative for the participating countries and our data indicate that cross-cultural validity of some of the scales is questionable. However, to expand on the analyses, one comparison between the participating nations’ was added on:  we reported how much COVID-19 impacted adolescents on a VAS scale between 0 and 10. Please see: Lines 113-120, New Figure 2, Lines 273-280 of results and 416-463 of discussion.

In addition, we would like to emphasize the importance and benefits of keeping a culturally diverse sample for analysis of gender differences.

The manuscript was language edited by an official language editing firm

Reviewer 5 Report

Paper deals with adolescents and theri react on the COVID-19 pandemic. Authors collected 5341 complete responses from five countries during zear of 2020 and 2021.The goal of the paper was to describe the self-rated impact of the COVID-19 pandemic’s restrictions on adolescents behavior, mood, relationships, and victimization.

The English language is appropriate and understandable. The article is well written and analyzes is presented appropriately. The research question is original and the results are interpreted appropriately. Nevertheless, I am of the opinion that the hypotheses of the paper should be more clearly emphasized in the text itself and put in abstract as well.

I also think that more theoretical sources should be added within Theoretical background part and as a separate chapter it should be added to the paper.

Author Response

Thank you for your time and recommendations to improve our paper.  We acknowledge the absence of clearly stated hypotheses. We added our hypotheses to the introduction section (Lines 110-127). Presentation of the hypotheses also improved our discussion.

We provided more theoretical background (Lines 98-105, 113-120 and 123-127) citing recently published (during the past month) reviews and meta-analyses on the subject.

The revised manuscript was sent to an official language editing firm. 

Round 2

Reviewer 1 Report

Dear Authors, 

I believe that this work has received a great deal of benefit from all comments and suggestions of the editors and I would suggest the editor to take this work in consideration for publication. The authors replied to all concerns in an exhaustive way.

Best,

Author Response

Thank you for your time and your constructive criticism. 

Reviewer 3 Report

My comments and concerns are appropriately addressed.  The paper is much improved.

Author Response

(The authors gave the same response as above.)

Reviewer 4 Report

Thank you for the opportunity to review the revised version of the paper. I totally see some improvement, but in the present form, I don’t see any overall merit.

I think that your research is very interesting, concerning a topic with high impact on adolescents’ life. There is a pretty extensive sample that gives the option to perform many advanced statistical analyses. Yet the have decided to present merely descriptive statistics, even after the first review round.

With such large samples, the representability per country is not such a great issue to not test for cross-cultural differences. Moreover, the authors replied that it is important to keep “a culturally diverse sample for analysis of gender differences”. Yet they can’t address this issue without conducting separate tests for gender differences per country. There is not even a chi-square test presented.

In the authors’ reply, they have implied that you have conducted validity tests per country of origin, and that the validity is questionable in some scales. This explains the low reliability presented in the paper.

As far as the behavioral change concerns, just percentages were presented, without addressing anything further. In some cases, there was not a change at all (e.g., 5% increased smoking while 5% decreased, same for Spending time with family taking part in fun/quality activities.

Figures: There should really have been added the percentages for each category in the figures that presented, as most of the data depicted are not presented elsewhere.

For all the above reasons and taking into consideration a) the status of IJERPH, and b) that this is a revised version, I suggest that it should be rejected, because despite the interesting topic, the paper itself adds little to the previous knowledge of adolescent psychology.

Author Response

Thank you for your time and your criticism.